# Agent0-VL: Exploring Self-Evolving Agent for Tool-Integrated Vision-Language Reasoning

**Jiaqi Liu**[1] **Kaiwen Xiong**[1] **Peng Xia**[1] **Yiyang Zhou**[1] **Haonian Ji**[1]
**Lu Feng**[1] **Siwei Han**[1] **Mingyu Ding**[1] **Huaxiu Yao**[1]

## Abstract

Large Vision-Language Models (LVLMs) have achieved remarkable progress in multimodal reasoning tasks; however, their learning remains constrained by the limitations of human-annotated supervision. Recent self-rewarding approaches attempt to overcome this constraint by allowing models to act as their own critics or reward providers. Yet, purely text-based self-evaluation struggles to verify complex visual reasoning steps and often suffers from evaluation hallucinations. To address these challenges, inspired by recent advances in tool-integrated reasoning, we propose Agent0-VL, a self-evolving vision-language agent that achieves continual improvement with tool-integrated reasoning. Agent0-VL incorporates tool usage not only into reasoning but also into self-evaluation and self-repair, enabling the model to introspect, verify, and refine its reasoning through evidence-grounded analysis. It unifies two synergistic roles within a single LVLM: a Solver that performs multi-turn tool-integrated reasoning, and a Verifier that generates structured feedback and fine-grained self-rewards through tool-grounded critique. These roles interact through a Self-Evolving Reasoning Cycle, where tool-based verification and reinforcement learning jointly align the reasoning and evaluation distributions for stable self-improvement. Through this zero-external-reward evolution, Agent0-VL aligns its reasoning and verification behaviors without any human annotation or external reward models, achieving continual self-improvement. Experiments on chart reasoning, geometric problem solving, and visual scientific analysis show that Agent0-VL achieves an 12.5% improvement over the base model.

## 1. Introduction

Vision-language agents have shown remarkable capabilities in tackling complex multimodal tasks (Wang et al., 2025a), including robotic manipulation (Intelligence et al.), visual question reasoning (Zhou et al., 2024a), and scientific discovery (Liu et al., 2025). Currently, most vision-language agents and Vision-Language Models (VLMs) are trained using human-annotated preference data (Huang et al., 2025; Su et al., 2025c; Wang et al., 2025c) or external-reward signals. However, such training paradigms are inherently constrained by the limitations of human annotators' preferences and the sparsity or incompleteness of environment-generated feedback, ultimately capping the upper bound of the agent's capabilities (Wang et al., 2025b;a). To enable agents to move beyond static human supervision and achieve continuous *self-evolution*, recent research has explored *self-rewarding learning* (Zhou et al., 2024b; Xiong et al., 2025), in which the agent or model itself acts as a Critic or Reward Model, providing feedback and reward signals for its own learning (Ding & Zhang, 2025; Wang et al., 2025b).

Nevertheless, for many complex visual reasoning tasks, purely text-based self-evaluation faces two key limitations. First, *limited evaluation capability*: when relying solely on textual reflection, models struggle to verify complex multi-step computations, spatial reasoning, or precise physical and geometric calculations, abilities that are critical for tasks such as visual geometry and chart analysis (Xu et al., 2025). Second, *unreliable evaluation process*: excessive textual reasoning causes models to rely on language shortcuts, bypassing fine-grained visual understanding and depending instead on linguistic priors or contextual bias (Zhou et al., 2024b; Li et al., 2025c). This often leads to evaluation hallucination, where the model incorrectly rewards a linguistically plausible yet visually incorrect answer, or penalizes a visually correct answer that fails to align with its language-based expectations.

To address these challenges, inspired by recent advances in

over the base model.

[1]UNC-Chapel Hill. Correspondence to: Huaxiu Yao <huaxiu@cs.unc.edu>.

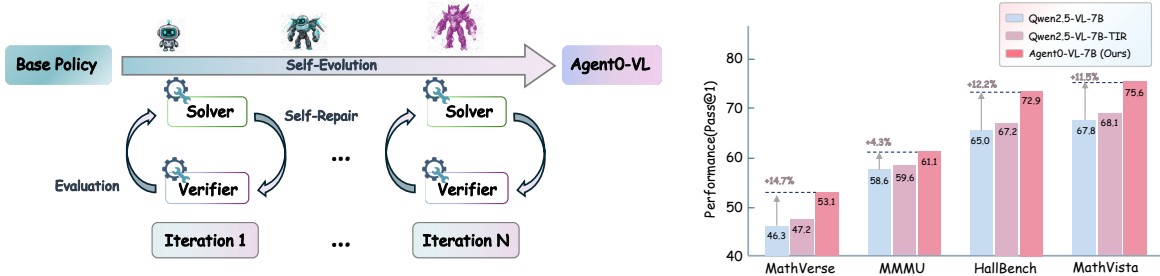

*Figure 1.* The evolve loop of Agent0-VL and its performance comparison. The left part illustrates the iterative evolution between the Solver and Verifier, where the Solver progressively refines reasoning strategies under Verifier feedback. The right part presents results showing that Agent0-VL outperforms tool-integrated reasoning methods across multiple representative benchmarks. *TIR*: Tool-Integrated Reasoning.

*tool-integrated reasoning* (Zhang et al., 2025d; Geng et al., 2025; Su et al., 2025c; Xue et al., 2025), we propose a simple yet effective idea: enable the model to employ external tools not only during reasoning, but also during its own *self-evaluation* and *self-repair*. By doing so, the model can perform closed-loop self-improvement under zero external reward supervision, learning to analyze, critique, and refine its reasoning in a verifiable manner. Building on this idea, we present **Agent0-VL**, a self-evolving vision-language agent framework that integrates tool-use paradigms into both the reasoning and self-evaluation processes. As illustrated in Figure 1, Agent0-VL unifies reasoning, verification, and self-repair within a single LVLM, which alternates between two synergistic roles: (1) the **Solver**, which performs multi-turn reasoning and selectively invokes external tools for grounded computation and visual perception; and (2) the **Verifier**, which validates intermediate reasoning steps through generative critique and tool-based feedback, generating fine-grained reward signals and repair instructions.

By alternating between these roles, the model forms a closed feedback loop that supports continual self-improvement. We formalize this iterative learning paradigm as a Self-Evolving Reasoning Cycle (SERC). In the *inner loop*, the model performs multi-turn reasoning, tool-based verification, and selective repair to progressively refine its reasoning trajectory. In the *outer loop*, we employ reinforcement learning, specifically, Group Relative Policy Optimization (GRPO) (Shao et al., 2024), to update the shared policy, jointly enhancing reasoning and verification capabilities over time. This process transforms the learning objective from static reward maximization into a process of distributional self-consistency, where reasoning and evaluation behaviors are jointly aligned and continually optimized.

Our main contribution is Agent0-VL, a unified self-evolving LVLM that integrates reasoning, verification, and self-repair into a single model trained entirely from zero external rewards. Experiments on geometric reasoning and visual scientific analysis demonstrate that Agent0-VL-7B achieves stable multi-iteration performance improvement, showing an average 12.5% improvement over the Qwen-VL base model. Furthermore, when used independently as a process reward model, Agent0-VL also improves the model's test-time scaling performance by an average of 7.3%.

## 2. Related Work

**Self-Evolving Methods.** To reduce the reliance on costly human supervision, recent research in both LLMs and VLMs has explored *self-evolving* techniques that enable models to improve autonomously using unlabeled data and self-generated feedback (Xia et al., 2025). In LLMs, a common strategy is to derive pseudo-rewards from the model's own outputs. For example, *self-consistency* leverages agreement among multiple generated solutions as a learning signal (Wang et al., 2022), which has enhanced reasoning in tasks like math and code generation, as seen in TTRL (Zuo et al., 2025), MM-UPT (Wei et al., 2025), and SRT (Shafayat et al., 2025). MM-UPT further improves spatial reasoning by generating synthetic geometry problems. Other approaches use a strong model as an automated evaluator (Pang et al., 2024), or rely on internal cues such as confidence scores (Zhao et al., 2025; Li et al., 2025a) and output entropy (Zhang et al., 2025c). DeepConf (Fu et al., 2025), for instance, enhances reasoning efficiency by selecting responses based on token entropy.

In the VLM domain, self-evolving agents are also progressing rapidly. ViPER (Zhang et al., 2025b) proposes a two-stage coarse-to-fine training framework that integrates instance- and image-level reconstruction with self-critiquing and prediction. In contrast, Vision-Zero (Wang et al., 2025b) employs a domain-agnostic, game-based self-play strategy. By alternating between self-play and reinforcement learning with verifiable rewards, it achieves scalable and sustained improvement. These methods demonstrate that VLMs, like LLMs, can evolve through structured feedback loops involving uncertainty modeling, task generation, and curriculum-based learning. In comparison, our work builds on these advances by introducing a tool-integrated visual reasoning and self-evaluation framework, enabling continual and stable performance improvements.

**Tool-Integrated Reasoning.** Tool-Integrated Reasoning (TIR) aims to enable models to invoke external tools (e.g., search engines (OpenAI, 2025b; Google, 2024; Team et al., 2025), calculators (Gou et al., 2023; Wang et al., 2023)) to overcome knowledge limitations and execute complex tasks (Schick et al., 2023; Yao et al., 2023; Guo et al., 2024; Zhou et al., 2025b;a; Geng et al., 2025). Subsequent work significantly enhanced LLM capabilities in multi-tool coordination, planning, and execution through instruction-tuning and agentic frameworks (Qin et al., 2023; Patil et al., 2023; Jin et al., 2025; Li et al., 2025b). Recent research began to extend TIR to VLMs, enabling models to not only process text but also dynamically interact with visual information. One line of work positions the VLM as an orchestrator that calls upon external vision expert tools (Yang et al., 2023; Hu et al., 2024) or skill repositories (Liu et al., 2024). Another line of work explores how VLMs can perform dynamic reasoning at the "vision level," treating image exploration itself as a tool, like zooming (Shen et al., 2025) and visual querying (Wu et al., 2025b). Recently, many efforts have adopted Reinforcement Learning (RL) for training (Su et al., 2025c;b;a). GRIT (Fan et al., 2025) and DeepEyes (Zheng et al., 2025) leverage RL to incentivize the model to actively cite image regions (e.g., bounding boxes) within its reasoning chains. Also some works (e.g., WebWatcher (Geng et al., 2025)) utilize RL to enhance the generalization of tool use for multimodal agents in deep research tasks (Narayan et al., 2025; Wu et al., 2025a). Building on this line of work, our method further integrates tool-augmented reasoning into the model's self-evaluation process, enabling the generation of process-level reward signals that guide more effective self-improvement.

# 3. Preliminaries

Multi-turn Tool-Integrated Reasoning (TIR) agents trained via Reinforcement Learning (RL) present substantial challenges, primarily due to the inherently compositional nature of reasoning and the partial observability of multimodal environments. To address these complexities, we formulate the reasoning process of our agent within the framework of a *partially observable Markov decision process (POMDP)*: $\mathcal{M} = (\mathcal{S}, \mathcal{A}, \mathcal{O}, T, R, \gamma)$, where $\gamma$ is the discount factor, $R$ is the reward function, which will be introduced in section 4. The whole multimodal reasoning dynamics and tool feedback are explicitly modeled through the following components:

**State Space.** $s_t \in \mathcal{S}$ denotes the latent multimodal reasoning state, encoding the textual reasoning context, visual features, and past tool input and output traces.

**Action Space.** Each action $a_t \in \mathcal{A}$ can be either: (1) a textual reasoning step $a_t^{(\text{text})}$, or (2) a structured tool invocation

$a_t^{(\text{tool})}$, which executes an external program, such as Python code. Hence, $\mathcal{A} = \mathcal{A}_{\text{text}} \cup \mathcal{A}_{\text{tool}}$.

**Observation Space.** An observation $o_t \in \mathcal{O}$ contains feedback from external tools or environment, such as returned numerical results or textual retrievals. The transition function $T(s_{t+1}|s_t, a_t, o_t)$ captures how state evolves given generated response and its corresponding tool feedback.

**Trajectory.** Given an input $x = (I, q)$, the model generates a multimodal trajectory: $\tau = \{(s_1, a_1, o_1), (s_2, a_2, o_2), \ldots, (s_T, a_T, o_T)\}$, where each transition encodes one reasoning-tool-feedback interaction.

# 4. Methodology

In this section, we introduce Agent0-VL, a self-evolving vision-language agent that integrates tool-use paradigms into both the reasoning and self-evaluation processes. The core idea is to enable a VLM to not only reason and solve problems, but also to verify, critique, and repair its own reasoning trajectories through a unified, self-evolving loop. We first describe the dual-role architecture composed of a *Solver* and a *Verifier*, then present how the model performs tool-grounded reasoning, verification, and confidence-gated self-repair. Finally, we detail the Self-Evolving Reasoning Cycle (SERC), where these roles interact through inner and outer optimization loops to achieve continual self-improvement.

## 4.1. Unified Solver-Verifier Architecture

To achieve autonomous evolution, as shown in Figure 2, Agent0-VL adopts a unified dual-role design, where a single LVLM alternates between two internal modes: a *Solver* that performs tool-integrated reasoning, and a *Verifier* that introspectively evaluates, critiques, and repairs the Solver's outputs. We detail the architecture as follows:

**Unified Policy Formulation.** We define a shared policy $\pi_\theta$ that governs both roles through a role indicator $m \in \{S, V\}$:

$$\pi_\theta(a_t|s_t, m) = \begin{cases} \pi_\theta^{\text{S}}(a_t|s_t), & m = \text{Solver}, \\ \pi_\theta^{\text{V}}(a_t|s_t, a_t, o_t), & m = \text{Verifier}, \end{cases} \quad (1)$$

where $s_t$ denotes the multimodal state, $a_t$ the generated reasoning action, and $o_t$ the tool or environment feedback.

**Solver.** When $m = S$, the model performs multi-turn reasoning and selectively invokes external tools. The resulting observations $o_t$ are incorporated into the context, allowing the agent to iteratively refine its reasoning with grounded evidence. Formally, the Solver follows

$$a_t \sim \pi_\theta^{\text{S}}(a_t|s_t), \quad b_{t+1} = f(b_t, o_t),$$

where $b_t$ represents the latent belief state encoding accumulated reasoning context and multimodal information.

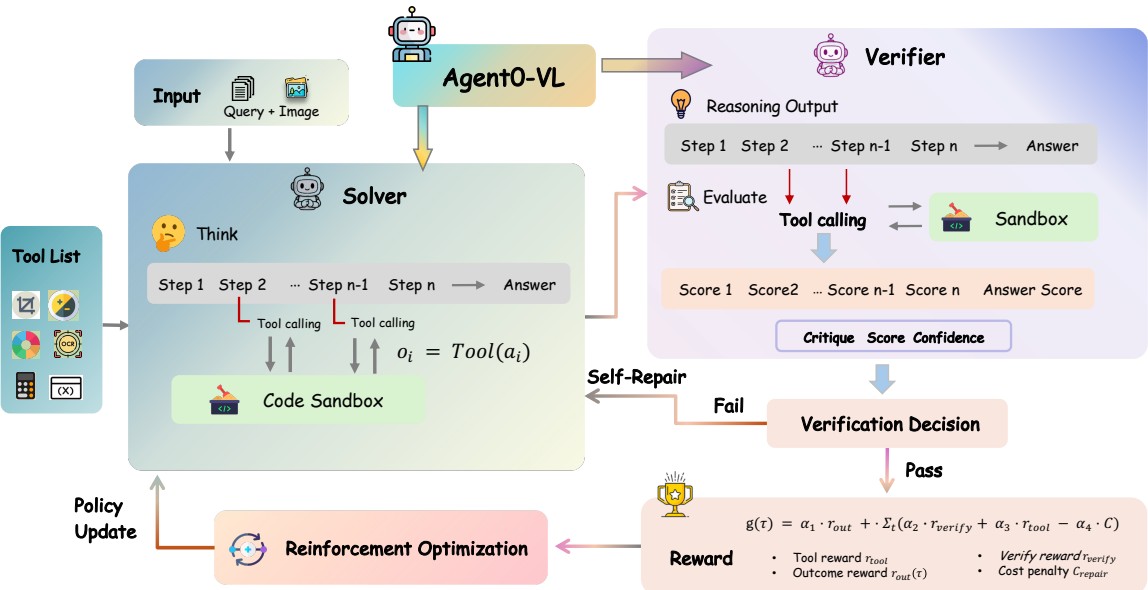

*Figure 2.* **The Framework of Agent0-VL.** The unified policy $\pi_\theta$ alternates between two internal roles: the **Solver** that generates reasoning trajectories with tool calls, and the **Verifier** that performs generative verification using tool feedback to produce critiques and step-wise rewards. These roles are jointly optimized through the Self-Evolving Reasoning Cycle, where self-generated rewards guide policy updates via RL.

**Verifier.** When $m = \mathrm{V}$, the model switches into a generative verification mode to evaluate each reasoning step. Given the triplet $(s_t, a_t, o_t)$, the Verifier outputs a feedback tuple:

$$V_t = (\mathrm{score}_t, \mathrm{conf}_t, \mathrm{critique}_t),$$

where $\mathrm{score}_t \in [-1, 1]$ measures factual correctness, $\mathrm{conf}_t \in [0, 1]$ estimates epistemic certainty, and $\mathrm{critique}_t$ provides natural-language reflection describing potential reasoning flaws. The Verifier can also re-invoke tools to cross-check correctness, providing a grounded and interpretable signal for self-evaluation.

### 4.2. Tool-Grounded Verification and Self-Repair

Within the dual-role framework, the *Solver* performs multi-turn reasoning by invoking external tools, while the *Verifier* provides process-level feedback and evaluation based on the Solver's reasoning trajectory and outputs. If the Verifier identifies deficiencies or inconsistencies in the Solver's reasoning, it initiates a *self-repair* process that leverages self-evaluation signals to revise and refine the reasoning trajectory, thereby improving reasoning accuracy.

Specifically, Agent0-VL introduces a structured *tool-grounded generative verification* mechanism designed to produce dense and interpretable feedback signals for reinforcement learning. At each step $t$, the Verifier produces $V_t$. During verification, the Verifier can query external tools to obtain factual evidence. By combining language-based reflection with executable tool-derived evidence, the model transforms verification from a static correctness check into a dynamic evaluation procedure, enabling iterative refinement

of its reasoning trajectories.

Based on the Verifier's step-wise assessments, we further design a corresponding *process-level reward* that integrates semantic reliability, tool-based validation, and cross-role regularization.

$$r_{\mathrm{proc}}^{(t)} = \lambda_{\mathrm{tool}} \cdot r(\mathrm{tool}_t) + \underbrace{\mathrm{score}_t \cdot \mathrm{conf}_t}_{\text{semantic reliability}} \tag{2}$$
$$- \beta_{\mathrm{div}} \cdot D_{\mathrm{KL}}\big(\pi_\theta^{\mathrm{V}} \| \pi_\theta^{\mathrm{E}}\big),$$

where $\lambda_{\mathrm{tool}}$ scales tool-based correctness, and $\beta_{\mathrm{div}}$ stabilizes the dual-role distributional alignment.

After completing self-verification, the *Verifier* determines whether the model should perform *self-repair* to correct its reasoning process based on the confidence score $\mathrm{conf}_t$ obtained during verification. Let $\tau_c$ denote a confidence threshold. The repair gate at step $t$ is defined as:

$$g_t = \sigma\big(\kappa(\tau_c - \mathrm{conf}_t)\big), \tag{3}$$

where $\sigma(\cdot)$ is the sigmoid function, and $\kappa$ controls the gating temperature. When $g_t$ is activated, the Verifier issues a local repair instruction $\Delta_t = f_\theta(s_t, a_t, V_t)$, and the Solver regenerates a corrected segment $a_t' \sim \pi_\theta(\cdot \mid s_t, \Delta_t, m = \mathrm{S})$.

Based on the above verification and self-repair processes, the resulting step-wise reward is then defined as:

$$r_t = r_{\mathrm{proc}}^{(t)} - g_t \cdot C_{\mathrm{repair}}^{(t)}, \tag{4}$$

where $C_{\mathrm{repair}}^{(t)}$ penalizes unnecessary repairs.

**Algorithm 1 Agent0-VL Training Process**

---

**Require:** Unified policy $\pi_\theta$, reference policy $\pi_{\theta_{\text{old}}}$, confidence threshold $\tau_c$.

1: Initialize $\pi_\theta$ with supervised data to learn tool usage and verification formats.
2: **for** each iteration $k = 1, \ldots, N_{\text{iter}}$ **do**
3:     **// Inner Loop: Generation and Self-Evaluation**
4:     **for** each task $x_i = (I_i, q_i)$ **do**
5:         Initialize trajectory $\tau_i \leftarrow \emptyset$, state $s_1$.
6:         **for** $t = 1, \ldots, T$ **do**
7:             Sample action $a_t \sim \pi_\theta(\cdot|s_t, m = \text{S})$ and execute to get $o_t$.
8:             Generate verification $V_t$.
9:             Compute process reward $r_{\text{proc}}^{(t)}$ (Eq. 2).
10:            **if** $\text{conf}_t < \tau_c$ **then**
11:                Generate repair $\Delta_t = f_\theta(s_t, a_t, V_t)$; resample $a_t' \sim \pi_\theta(\cdot|s_t, \Delta_t, \text{RA})$.
12:            **end if**
13:            Compute effective step reward(Eq. 4).
14:         **end for**
15:         Compute total trajectory return $g(\tau_i)$ (Eq. 5).
16:     **end for**
17:     **// Outer Loop: Policy Update**
18:     Optimize $\pi_\theta$ via GRPO using all collected trajectories (Eq. 6).
19: **end for**

---

### 4.3. Self-Evolving Reasoning Cycle

Having defined the agent's architecture and verification mechanisms, we now introduce the model training process (see Algorithm 1 for the whole process). Specifically, the training process is divided into two stages. First, a supervised fine-tuning (SFT) stage for cold-start initialization, ensuring the model learns proper tool usage and self-evaluation formats. Second, a RL-based self-evolving reasoning cycle (SERC) that operationalizes this architecture, enabling the agent to learn from its own tool-grounded feedback. SERC consists of two nested loops: an inner loop for data generation and an outer loop for policy evolution.

The inner loop is responsible for generating experience by coupling reasoning and verification. For a given multimodal task $x$, the *Solver* first generates a complete reasoning trajectory $\tau = \{(s_t, a_t, o_t)\}_{t=1}^T$, invoking external tools as needed. Immediately following this, the *Verifier* re-evaluates each step $t$ of the trajectory to produce the verification tuple $V_t$ and the corresponding process reward $r_{\text{proc}}^{(t)}$ (using Eq. 2). If the Verifier's confidence for a step is below the threshold $\tau_c$, the selective repair mechanism (Eq. 4) is triggered, and a cost $C_{\text{repair}}$ is applied. Finally, the total return $g(\tau)$ for the trajectory is aggregated, integrating both the final outcome

reward $r_{\text{out}}$ and the accumulated step-wise process rewards:

$$g(\tau) = \alpha_{\text{out}}\, r_{\text{out}} + \sum_{t=1}^{T} \gamma^{t-1} r_t, \tag{5}$$

where $\alpha_{\text{out}}$ balances the two reward types and $\gamma$ is the discount factor.

The outer loop then uses these generated trajectories to evolve the unified policy $\pi_\theta$. We employ Group Relative Policy Optimization (GRPO) (Shao et al., 2024), a variant of PPO designed for generative tasks. For a group of $G$ trajectories $\{\tau_i\}_{i=1}^G$ sampled under the current policy, we compute a normalized advantage for each trajectory:

$$\hat{A}_i = \frac{g(\tau_i) - \text{mean}(g_1, \ldots, g_G)}{\text{std}(g_1, \ldots, g_G) + \varepsilon},$$

where $g_i = g(\tau_i)$ and $\varepsilon$ is a normalization constant. This relative advantage $\hat{A}_i$ reframes the learning objective as "being better than the group average." The policy is then optimized to increase the likelihood of trajectories with positive advantages while maintaining stability through a KL regularization term:

$$\begin{aligned}
\mathcal{L}_{\text{EDLP}} = -\mathbb{E}_i &\Big[ \min\Big( \rho_i \hat{A}_i,\ \text{clip}(\rho_i, 1 - \epsilon, 1 + \epsilon)\hat{A}_i \Big) \Big] \\
&+ \beta_{\text{KL}}\, D_{\text{KL}}\big(\pi_\theta \| \pi_{\theta_{\text{old}}}\big),
\end{aligned} \tag{6}$$

where $\rho_i = \frac{\pi_\theta(\tau_i)}{\pi_{\theta_{\text{old}}}(\tau_i)}$ is the importance sampling ratio.

## 5. Experiments

In this section, we evaluate Agent0-VL across multiple visual reasoning benchmarks to answer the following questions: (1) Can it improve reasoning ability over existing open-source baselines? (2) Does repeated self-evolution through multiple iterations yield consistent performance gains? (3) How effective are the proposed components? and (4) Can Agent0-VL serve as a process reward model to enhance other LVLMs?

### 5.1. Experimental Setup

**Benchmarks.** To evaluate the effectiveness of our method, we conducted experiments on a set of seven benchmarks. The science and mathematical benchmarks include MathVerse (Zhang et al., 2024a), MathVision (Wang et al., 2024), MathVista (Lu et al., 2023), WeMath (Qiao et al., 2024), and MMMU (Yue et al., 2024), while other benchmarks consist of HallusionBench (Guan et al., 2024), and ChartQA (Masry et al., 2022).

**Baselines.** Additionally, we compared Agent0-VL against three types of baselines: (1) Closed-Source LVLMs, including GPT-4o (Achiam et al., 2024), o1 (OpenAI,

*Table 1.* Comparison of model performance across representative visual reasoning benchmarks. Closed-source LVLMs are listed for reference, while open-source models are grouped by general-purpose and reasoning-oriented categories. The best results among all open-sourced models are **bolded** and the second best results are underlined in the table. *Note:* Qwen2.5-VL-7B-TIR and Qwen3-VL-8B-TIR denote the results of integrating tool-enhanced reasoning into the corresponding base models, obtained from our local fine-tuning and evaluation.

| Model | MathVerse | MathVision | MathVista | WeMath | HallBench | ChartQA | MMMU | Avg. |
|---|---|---|---|---|---|---|---|---|
| *Close-source MLLMs* | | | | | | | | |
| GPT-4o (Achiam et al., 2024) | 50.8 | 30.4 | 63.8 | 68.8 | 55.0 | 85.7 | 69.1 | 60.5 |
| OpenAI-o1 (OpenAI, 2024) | 57.0 | 60.3 | 73.9 | - | - | 83.1 | 77.6 | - |
| Claude-3.7-Sonnet (Anthropic, 2025) | 52.0 | 41.3 | 66.8 | 72.6 | 55.4 | 56.5 | 75.0 | 59.9 |
| *Open-Source General MLLMs* | | | | | | | | |
| InternVL-2.5-8B (Chen et al., 2025b) | 39.5 | 19.7 | 64.4 | 53.5 | 61.7 | 79.1 | 62.7 | 54.4 |
| InternVL-3-8B (Zhu et al., 2025) | 39.8 | 29.3 | 71.6 | 58.1 | 64.3 | 85.9 | 60.7 | 58.5 |
| Qwen2.5-VL-7B (Bai et al., 2025) | 46.3 | 25.1 | 67.8 | 62.1 | 65.0 | 83.5 | 58.6 | 58.3 |
| Qwen2.5-VL-7B-TIR* | 47.2 | 26.3 | 68.1 | 63.7 | 67.2 | 84.1 | 59.6 | 59.5 |
| Qwen3-VL-8B (Team, 2025) | 62.1 | 53.9 | 77.2 | 72.5 | 72.1 | 84.6 | 69.6 | 70.3 |
| Qwen3-VL-8B-TIR* | 63.1 | 54.7 | 79.4 | 73.1 | 72.8 | 85.4 | 70.9 | 71.3 |
| *Open-Source Reasoning MLLMs* | | | | | | | | |
| R1-VL-7B (Zhang et al., 2025a) | 40.4 | 24.7 | 63.5 | 60.1 | 54.7 | 76.1 | - | - |
| Vision-R1-7B (Huang et al., 2025) | 51.9 | 30.7 | 73.5 | 73.9 | 68.8 | 79.8 | 50.5 | 61.3 |
| R1-OneVision-7B (Yang et al., 2025) | 46.4 | 29.9 | 64.1 | 61.8 | 67.5 | 77.8 | - | - |
| OpenVLThinker-7B (Deng et al., 2025) | 45.7 | 26.3 | 71.2 | 66.7 | 70.2 | 78.4 | - | - |
| VLAA-Thinker-7B (Chen et al., 2025a) | 52.7 | 29.2 | 69.7 | 70.2 | 68.2 | 80.1 | - | - |
| MM-Eureka-Qwen-7B (Meng et al., 2025) | 50.5 | 27.9 | 73.6 | 67.4 | 66.9 | 82.1 | 52.7 | 60.2 |
| ThinkLite-VL-7B (Wang et al., 2025d) | 52.1 | 32.9 | 75.1 | 69.3 | 70.9 | 84.8 | 55.5 | 62.9 |
| Thyme-VL-7B (Zhang et al., 2025d) | 51.3 | 27.6 | 70.0 | - | 71.0 | 86.1 | - | - |
| **Agent0-VL-7B (Ours)** | 53.1 | 37.3 | 75.6 | 71.7 | 72.9 | 87.3 | 61.1 | 65.6 |
| **Agent0-VL-8B (Ours)** | **65.5** | **56.2** | **83.7** | **79.6** | **74.3** | **89.7** | **73.4** | **74.6** |

2024), Gemini-2.0-pro (Deepmind, 2025), and Claude-3.7-Sonnet(Anthropic, 2025); (2) Open-Source General MLLMs, including Qwen2.5-VL-3B (Bai et al., 2025), Qwen2.5-VL-7B (Bai et al., 2025), Qwen2.5-VL-32B (Bai et al., 2025), InternVL-2.5-8B (Chen et al., 2025b) and InternVL3-8B (Zhu et al., 2025); and (3) Open-Source Reasoning MLLMs, including R1-VL-7B (Zhang et al., 2025a),Vision-R1-7B (Huang et al., 2025), R1-Onevision-7B (Yang et al., 2025), OpenVLThinker-7B (Deng et al., 2025), VLAA-Thinker-7B (Chen et al., 2025a), MM-Eureka-7B (Meng et al., 2025), Thyme (Zhang et al., 2025d), Qwen3-VL-8B (Team, 2025), and ThinkLite-VL-7B (Wang et al., 2025d).

**Training Datasets.** We construct a large-scale multi-modal reasoning corpus tailored for both SFT and RL stages. The SFT dataset (200k samples) is built from open-source benchmarks including Geometry3K (Lu et al., 2021), GeoQA (Chen et al., 2022), Mulberry dataset (Yao et al., 2024), MM-Eureka (Meng et al., 2025), and Re-tool (Feng et al., 2025), where we automatically generate tool-augmented reasoning trajectories using GPT-5 and Qwen2.5-VL-72B as teacher models. For the RL stage, we construct an additional 40k dataset. Detailed data construction procedures are provided in Appendix D.

## 5.2. Main Results

**Overall Performance.** Table 1 summarizes the performance of Agent0-VL across all visual reasoning benchmarks. Our model consistently achieves substantial im-

provements over all open-source baselines. Specifically, Agent0-VL-7B outperforms existing open-source 7B models, achieving a 4.29% average gain over ThinkLite-VL-7B. Compared to the base Qwen2.5-VL-7B, it shows a 12.5% improvement, and a 10.3% gain over its tool-integrated variant Qwen2.5-VL-7B-TIR. Even with a stronger base model (Qwen3-VL-8B), Agent0-VL maintains strong compatibility and further improves performance, surpassing the base by 6.1% and its tool-augmented variant (Qwen3-VL-8B-TIR) by 4.6%. Notably, Agent0-VL-8B also outperforms closed-source systems such as GPT-4o on key benchmarks including MathVista, HallBench, and ChartQA, underscoring the generalization and reasoning strength of our approach.

**Performance Across Task Domains.** In domain-specific evaluations, Agent0-VL demonstrates the most significant improvements on mathematical reasoning benchmarks such as MathVista and WeMath, where tool-grounded execution and verification play a crucial role in accurate symbolic reasoning. Our 7B and 8B models achieve 18.1% and 7.4% improvements, respectively, over their base models on math-related benchmarks. For perception-heavy benchmarks such as HallusionBench and ChartQA, integrating the Verifier's factual grounding substantially reduces visual hallucinations, leading to 12.2% and 3.1% improvements compared to the base model. These results indicate that Agent0-VL enhances both symbolic reasoning and visual understanding, confirming the robustness and generality of our framework across diverse task domains.

**Iterative Self-Evolution.** We further investigate how the

*Table 2.* Performance comparison of Agent0-VL-7B across iterative training stages. Each iteration (*Iter 1-3*) progressively refines reasoning and tool-grounded capabilities, consistently outperforming the base model across all benchmarks.

| Model Name | MathVerse | MathVision | MathVista | WeMath | HallBench | ChartQA | MME-Real | MMMU | Avg. |
|---|---|---|---|---|---|---|---|---|---|
| Base Model | 46.3 | 25.1 | 67.8 | 62.1 | 65.0 | 83.5 | 58.3 | 50.6 | 57.3 |
| Iter 1 | 48.4 | 29.6 | 69.2 | 66.8 | 67.9 | 84.7 | 63.9 | 53.7 | 60.5 |
| Iter 2 | 51.1 | 35.3 | 72.8 | 70.1 | 70.3 | 86.1 | 64.7 | 58.3 | 63.6 |
| Iter 3 | 53.1 | 37.3 | 75.6 | 71.7 | 72.9 | 87.3 | 65.3 | 61.1 | 65.5 |

*Table 3.* Ablation study of Agent0-VL on different benchmarks. Removing *Self Repair*, *Tool-Grounded Reward*, or *SERC* notably degrades performance, highlighting the complementary roles of these modules.

| Setting | Math Avg. | HallBench | ChartQA | MME-Real | MMMU |
|---|---|---|---|---|---|
| Agent0-VL-7B | 59.4 | 72.9 | 87.3 | 65.3 | 61.1 |
| *w/o* Self Repair | 57.5 | 71.6 | 86.1 | 64.1 | 57.9 |
| *w/o* Tool Use | 53.1 | 67.5 | 86.2 | 61.9 | 54.7 |
| *w/o* SERC (SFT only) | 51.8 | 65.8 | 85.4 | 60.5 | 52.5 |
| Qwen2.5-VL-7B(Base Model) | 50.3 | 65.0 | 83.5 | 58.3 | 50.6 |

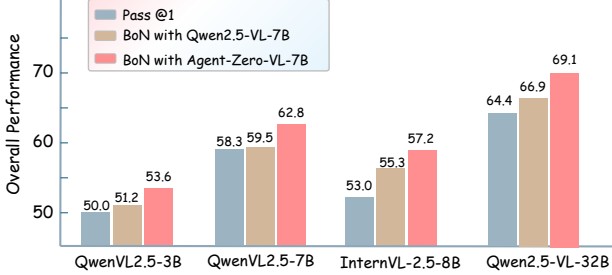

*Figure 3.* The overall Best-of-8 evaluation results across seven multimodal reasoning benchmarks with different critic models. Our model greatly enhances the overall performance compared with Qwen2.5-VL-7B model.

model's reasoning capabilities evolve across multiple iterations of SERC. As shown in Table 2, Agent0-VL demonstrates a steady and monotonic improvement over time: in the first iteration, its overall performance increases by 5.2% compared to the base model, followed by additional gains of 4.0% and 2.8% in the second and third iterations, respectively. These results validate the effectiveness of our self-evolving framework, showing that the model can achieve stable and continuous performance gains through iterative self-improvement.

### 5.3. Ablation Studies

To understand the contribution of each major component in Agent0-VL, we conduct controlled ablation experiments focusing on three key modules: (*i*) the SERC for reinforcement learning (*ii*) Tool Use, and (*iii*) Self Repair. For each ablation, we remove the corresponding module while keeping all other components intact, and retrain the model under the same settings. The results are summarized in Table 3.

We report the results in Table 3. According to the results,

first, we observe that eliminating the outer reinforcement learning loop (SERC) and relying solely on SFT training leads to the most significant performance drop, with an average decrease of 8.7% across all benchmarks. This result confirms that the bidirectional interaction between reasoning and evaluation is essential for enabling long-term self-evolution, which constitutes one of the core contributions of this work. Second, when tool usage is removed and the model performs only text-based reasoning and self-evaluation, the average performance declines by 6.5%. This highlights the crucial role of tool integration in enhancing reasoning accuracy, especially for mathematical and fact-verification tasks. Finally, removing the self-repair mechanism, allowing the model to self-evaluate without performing explicit correction, results in a moderate average performance drop of 2.5%. In particular, for mathematical reasoning benchmarks, re-executing reasoning through selective repair provides additional opportunities for resampling and correction, thereby improving overall reliability.

### 5.4. Performance as a Process Reward Model

To further evaluate the generalization and standalone utility of the Verifier module, we deploy it as a Process Reward Model (PRM) to assess reasoning trajectories generated by other VLMs. This experiment serves two key purposes: (*i*) to examine whether the Verifier can generalize beyond Agent0-VL and reliably evaluate the step-level correctness of external models' outputs, and (*ii*) to validate the effectiveness of our tool-grounded reward modeling approach, even when decoupled from policy learning.

As shown in Figure 3, our verifier significantly improves Best-of-8 (BoN) performance when used as a reward scorer across various LVLMs. Compared to Qwen2.5-VL-7B as the critic model, Agent0-VL consistently yields stronger trajectory selection across models of different scales, from

*Table 4.* Performance of the Agent0-VL as a PRM. "+Ours" denotes models integrated with the Agent0-VL for reward scoring. Across diverse multimodal reasoning benchmarks, our method consistently enhances accuracy, validating its effectiveness as a generalizable, tool-grounded reward model.

| Model | MathVerse | MathVision | MathVista | WeMath | HallBench | ChartQA | Overall |
|---|---|---|---|---|---|---|---|
| Qwen2.5-VL-3B (Bai et al., 2025) | 34.8 | 21.9 | 58.4 | 51.7 | 59.8 | 73.1 | 50.0 |
| +Ours | 38.9 | 26.1 | 65.8 | 54.2 | 61.2 | 75.2 | 53.6 |
| Qwen2.5-VL-7B (Bai et al., 2025) | 46.3 | 25.1 | 67.8 | 62.1 | 65.0 | 83.5 | 58.3 |
| +Ours | 51.2 | 33.6 | 72.3 | 66.9 | 68.1 | 84.6 | 62.8 |
| InternVL2.5-8B (Chen et al., 2025b) | 39.5 | 19.7 | 64.4 | 53.5 | 61.7 | 79.1 | 53.0 |
| +Ours | 43.2 | 26.2 | 67.3 | 59.8 | 63.6 | 83.0 | 57.2 |
| InternVL2.5-8B (Chen et al., 2025b) | 39.8 | 29.3 | 71.6 | 58.1 | 64.3 | 85.9 | 58.2 |
| +Ours | 45.4 | 33.0 | 74.6 | 62.5 | 67.2 | 88.3 | 61.8 |
| Qwen2.5-VL-32B (Bai et al., 2025) | 48.5 | 38.4 | 74.7 | 69.1 | 71.8 | 84.0 | 64.4 |
| +Ours | 53.0 | 44.3 | 78.6 | 75.2 | 74.7 | 88.5 | 69.1 |

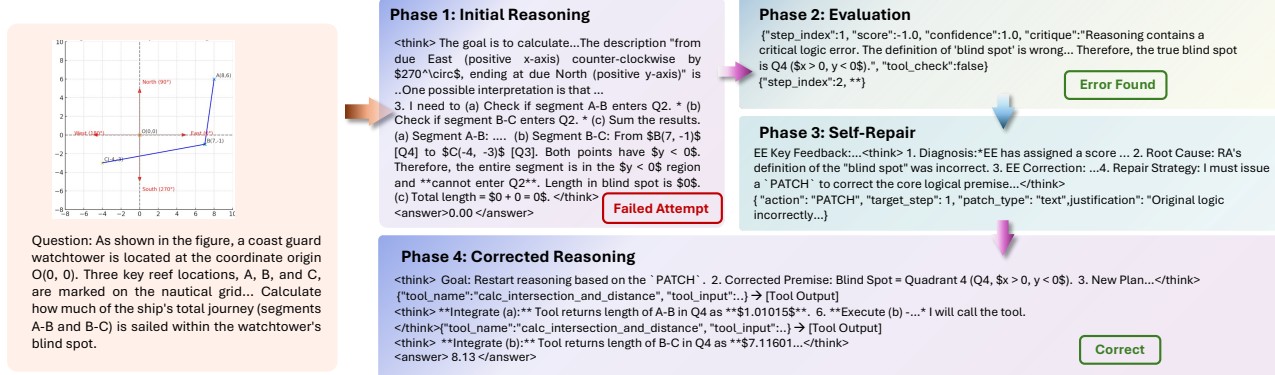

*Figure 4.* Simplified illustration of Agent0-VL's self-evolving reasoning process on a geometric reasoning task during training phase.

Qwen2.5-VL-3B up to Qwen2.5-VL-32B, demonstrating more effective reward assignment and sharper step-level discrimination. Table 4 further highlights the effectiveness of our model, showing substantial improvements in both step-level reward correlation and overall accuracy across seven multimodal reasoning benchmarks. When integrated as a PRM into different open-source LVLMs, Agent0-VL consistently enhances reasoning stability and factual grounding, yielding an average gain of 7.3%. Notably, even smaller models such as Qwen2.5-VL-3B benefit significantly from the structured feedback, indicating strong generalization across architectures and model scales.

These results validate that Agent0-VL not only improves its own reasoning through internal reward optimization, but also serves as a general-purpose, interpretable verifier capable of externally supervising or refining other reasoning models.

### 5.5. Case Study

We present a representative visual reasoning case (Figure 4) to illustrate the self-evolving reasoning behavior of Agent0-VL. Unlike conventional LVLMs that terminate after an incorrect prediction, Agent0-VL iteratively refines its rea-

soning through a structured self-evaluation and repair loop. In this example sampled during RL training, the *Solver* initially produces an incorrect answer due to a misinterpretation of the "blind spot" segment. The *Verifier* detects a contradiction via step-level assessment and tool invocation, and returns high-confidence feedback identifying the faulty step and its cause. The *Self-Repair* module then generates a corrective patch, which is re-injected into the reasoning chain, enabling the *Solver* to re-execute and reach the correct answer. A simplified view is shown in Figure 4, with the full multi-phase visualization provided in Appendix 5.5.

## 6. Conclusion

We presented Agent0-VL, a self-evolving vision–language agent that unifies reasoning, verification, and self-repair within a single model. Through its dual roles, the *Solver* and the *Verifier*, Agent0-VL operates under the Self-Evolving Reasoning Cycle, where the model continually refines its reasoning through tool-grounded verification, confidence-gated repair, and reinforcement learning. Empirically, Agent0-VL outperforms existing open-source models across a wide range of visual reasoning benchmarks, achieving stronger factual consistency and multi-turn stability.

## Impact Statement

This work aims to advance the field of machine learning. While it may have broader societal implications, we do not identify any specific impacts that require separate discussion at this time.

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

## A. Notation

## B. Implementation Details

**Base Model.** Agent0-VL is implemented upon Qwen2.5-VL-7B-Instruct and Qwen3-VL-8B. Both the Solver and Verifier share parameters $\theta$.

**Training Details.** Our training pipeline consists of two sequential stages: supervised fine-tuning (SFT) and RL. In the SFT stage, Agent0-VL is first trained on tool-usage and image manipulation data, followed by gradual annealing on mathematical code reasoning data. The learning rate is set to $1 \times 10^{-5}$, while other hyperparameters remain consistent. We adopt a batch size of 128, train for 3 epochs, and apply a linear warmup ratio of 0.05. To ensure training stability and prevent early-stage collapse or entropy degeneration in the RL process, we perform a short external-reward pretraining phase before self-evolution begins. Specifically, the model is trained for 2 epochs using conventional RL with externally defined correctness signals to initialize stable exploration and tool-usage behaviors. This pretraining serves as a warm-up that activates structured exploration, after which the model transitions to the self-evolving RL phase driven purely by internal process rewards. In the self-evolving RL stage, the learning rate is set to $5 \times 10^{-7}$, and training is performed for 1 epoch with a batch size of 256. The reinforcement learning follows the GRPO (Shao et al., 2024) paradigm with a group size $N = 8$ for relative normalization. We set the KL divergence coefficient $\beta_{\mathrm{KL}} = 0.001$ to regularize policy drift, collect 4 rollouts per task, and apply a repetition penalty of 1.05 to discourage redundant reasoning. The entropy coefficient is $\beta_{\mathrm{ent}} = 0.01$, the selective repair threshold $\tau_c = 0.7$, and the repair penalty $\eta = 0.05$. All experiments are conducted with mixed-precision training (bfloat16) on 8 NVIDIA H200 GPUs.

## C. Prompting and Templates

We provide system prompts for our framework, as shown in Figure 5-7, respectively.

## D. Training Data Construction Pipeline

This section describes the multi-stage data construction pipeline used to train Agent0-VL. The objective is to build a large-scale corpus of high-quality, tool-integrated reasoning trajectories that equip the model with both reasoning and verification capabilities before entering the self-evolving reinforcement stage.

### D.1. Overview

To ensure diversity and robustness, the training data are constructed through a progressive curriculum, covering a

---

**System Prompt(Solver)**

```
You are the Reasoner in a unified
    tool-integrated VLM agent. You
    must solve tasks through
    multi-turn reasoning and
    selective tool use.

Rules:
- Wrap internal reasoning in
    <think>...</think>.
- Call tools only when necessary
    using: {"tool_name":"<Tool>",
"tool_input":{...}}

- Wait for tool_output before
    continuing.
- Be concise and deterministic; no
    hallucinated values.

Process:
<think>
1. Restate goal & plan next step.
2. Decide if a tool is needed;
    specify input.
3. Integrate tool_output; update
    reasoning.
</think>
Emit one tool call or continue
    reasoning.

When finished:
(1) Output CONFIDENCE: <0-1>
(2) Output FINAL_ANSWER: <answer>.
```

*Figure 5.* System prompt for the Solver.

---

wide range of multimodal reasoning tasks. The resulting corpus is divided into three functional tiers:

- **Direct reasoning data:** single-turn textual or visual questions solvable without tool calls, grounding linguistic reasoning and visual perception.

- **Tool-augmented data:** problems that require code execution, OCR, or visual analysis, providing factual grounding and verifiable evidence.

- **Multi-turn reasoning data:** complex visual–mathematical tasks involving iterative tool use, correction, and reflection.

All prompts are derived or adapted from existing multimodal benchmarks, including Geometry3K (Lu et al., 2021), GeoQA (Chen et al., 2022), Mulberry (Yao

| Symbol | Meaning |
|---|---|
| $x$ | Task instance (image + text) |
| $s_t, a_t, o_t$ | State/context, action (reasoning/tool call), observation (tool output) at step $t$ |
| $\tau$ | Trajectory $\{(s_t, a_t, o_t)\}_{t=1}^T$ |
| $\pi_\theta$ | Unified policy with role tokens for RA/EE |
| $r_{\text{proc}}^{(t)}$ | Process-level reward at step $t$ |
| $g(\tau)$ | Composite trajectory reward (Eq. 5) |
| $\tau_c$ | Confidence threshold for triggering local repair |

*Table 5.* Key notations used throughout the paper.

et al., 2024), LLaVA-OV-Image (Li et al., 2024a), MM-RLHF (Zhang et al., 2025e), SMR (Zhang et al., 2024b), MM-Eureka (Meng et al., 2025), Retool (Feng et al., 2025), and arXivQA (Li et al., 2024b). For mathematical and chart-based reasoning, we further include MathVerse (Zhang et al., 2024a), MathVista (Lu et al., 2023), WeMath (Qiao et al., 2024), and ChartQA (Masry et al., 2022).

### D.2. Multi-Stage SFT Data Construction

The SFT stage aims to teach Agent0-VL to perform end-to-end, tool-integrated reasoning and self-verification under minimal human supervision. We employ a three-step automatic pipeline inspired by recent multimodal reasoning works (Meng et al., 2025; Zhang et al., 2025e).

**(1) Task Sampling and Prompt Construction.** We randomly sample multimodal problems $(Q, I)$ from the datasets above. For each problem, a structured prompt is applied that requests: (i) a detailed reasoning trace (wrapped in <think>...</think>) and (ii) explicit tool-use plans in JSON format. Teacher models (GPT-5 (OpenAI, 2025a) and Qwen2.5-VL-72B (Bai et al., 2025)) are used to bootstrap the generation of complete trajectories.

**(2) Tool Execution and Verification.** All tool calls are executed in a sandbox environment to obtain real results. For each reasoning step $a_t$, the corresponding observation $o_t$ is logged, forming complete multimodal trajectories $\tau = \{(a_t, o_t)\}_{t=1}^T$. An auxiliary verifier model then checks the consistency between textual reasoning, tool outputs, and the final answer. Inconsistent or unexecutable trajectories are automatically filtered.

**(3) Tool-Grounded Self-Verification and Repair Examples.** To initialize the model's self-evaluation ability, we further include examples where the verifier generates structured feedback $V_t = (\text{score}_t, \text{conf}_t, \text{critique}_t)$ and correction signals for low-confidence steps. These "reasoning–verification–repair" triples serve as bridge samples for transitioning from supervised learning to self-evolving reinforcement learning.

After filtering and deduplication, we obtain approximately 200k high-quality multimodal trajectories. This unified SFT dataset initializes the dual-role reasoning and verification behaviors of Agent0-VL, laying the foundation for self-evolving optimization.

### D.3. Dataset For Reinforcement Learning

We start from math and perception problems from MathVerse (Zhang et al., 2024a), MathVista (Lu et al., 2023), WeMath (Qiao et al., 2024), arXivQA (Li et al., 2024b), and ChartQA (Masry et al., 2022), and ThinkLiteVL (Wang et al., 2025d). The model rolls out reasoning trajectories under its current policy, while the internal Verifier generates step-level rewards and critiques to guide optimization.

### D.4. Quality Control

To ensure factual reliability and semantic alignment, we apply:

- **Execution validation:** all tool invocations are re-run in a sandbox to remove invalid traces.

- **Semantic consistency check:** textual reasoning must align with tool results and visual observations.

- **Redundancy filtering:** near-duplicate reasoning traces are pruned via embedding similarity.

- **Manual spot-check:** about 10k samples are manually reviewed for multimodal and reasoning correctness.

This systematic construction ensures that Agent0-VL learns not only to reason and act across modalities, but also to verify, repair, and self-evolve through grounded, tool-based evidence.

In total, Agent0-VL is trained on approximately 240k curated multimodal reasoning trajectories: ∼200k SFT trajectories for cold-start reasoning, and ∼40k data for RL.

```
You are the Verifier in a unified
    tool-integrated VLM agent. Your
    role is to verify a given
    reasoning trajectory
    step-by-step, optionally calling
    tools to check facts.

Inputs: trajectory prefix
    $\tau_{1:t} = {(s_k, a_k,
    o_k)}_{k=1..t}$.

Rules:

(1)Wrap internal thought in
    <think>...</think>.
(2)Use the same tool schema for
    factual checks.
(3) Output exactly one JSON line per
    step:
{"step_index":t,
 "score":<-1-1>,
 "confidence":<0-1>,
 "critique":"<<=2 sentences>",
 "tool_check":true|false
 }

Principles:

(1) Ground verification on objective
    tool evidence.
(2) Penalize unsupported or
    inconsistent reasoning.
(3) High confidence requires
    agreement between tool and text.
```

*Figure 6.* System prompt for the Verifier.

# E. Case Studies

To provide an intuitive understanding of how Agent0-VL performs reasoning, verification, and tool interaction in practice, we visualize several representative cases covering both single-step and multi-step reasoning scenarios. These examples illustrate how the model dynamically invokes external tools, integrates tool outputs into its internal reasoning process, and leverages the Verifier for process-level verification and correction. Together, these cases highlight the model's ability to ground reasoning in visual evidence, maintain factual consistency, and perform self-reflective improvement across diverse visual reasoning tasks.

```
You are the Self-Repair module of a
    unified tool-integrated VLM.
You receive: (i) the original
    trajectory prefix \tau_{1:t},
    (ii) the EE verification triple
    for step t (score_t,
    confidence_t, critique_t), and
    (iii) the minimal repair target
    (segment u^(t)).

GOAL
- If confidence_t < \tau_c, propose
    a minimal, local patch to u^(t)

    that fixes the specific error
    WITHOUT rewriting validated
    context.
- Use tools to recompute only what
    is necessary to validate the
    patch.

REASONING FORMAT
- Put your full planning and
    diagnostics inside
    <think>...</think>.
- After <think>, either (A) emit a
    PATCH JSON (and optionally a
    tool call), or (B) emit a
    NO_CHANGE JSON if repair is not

    warranted.

PATCH JSON (single line):
{
  "action": "PATCH",
  "target_step": t,
  "patch_type":
    "<text|code|tool_call|parameter>",
  "new_content": "<the minimal
    replacement content>",
  "justification": "<<= 2 sentences
    referencing critique/evidence>"
}

NO_CHANGE JSON (single line):

{
  "action": "NO_CHANGE",
  "target_step": t,
  "reason": "<why repair is not
    warranted or evidence is
    insufficient>"
}
```

*Figure 7.* System prompt for Self-Repair Mode.

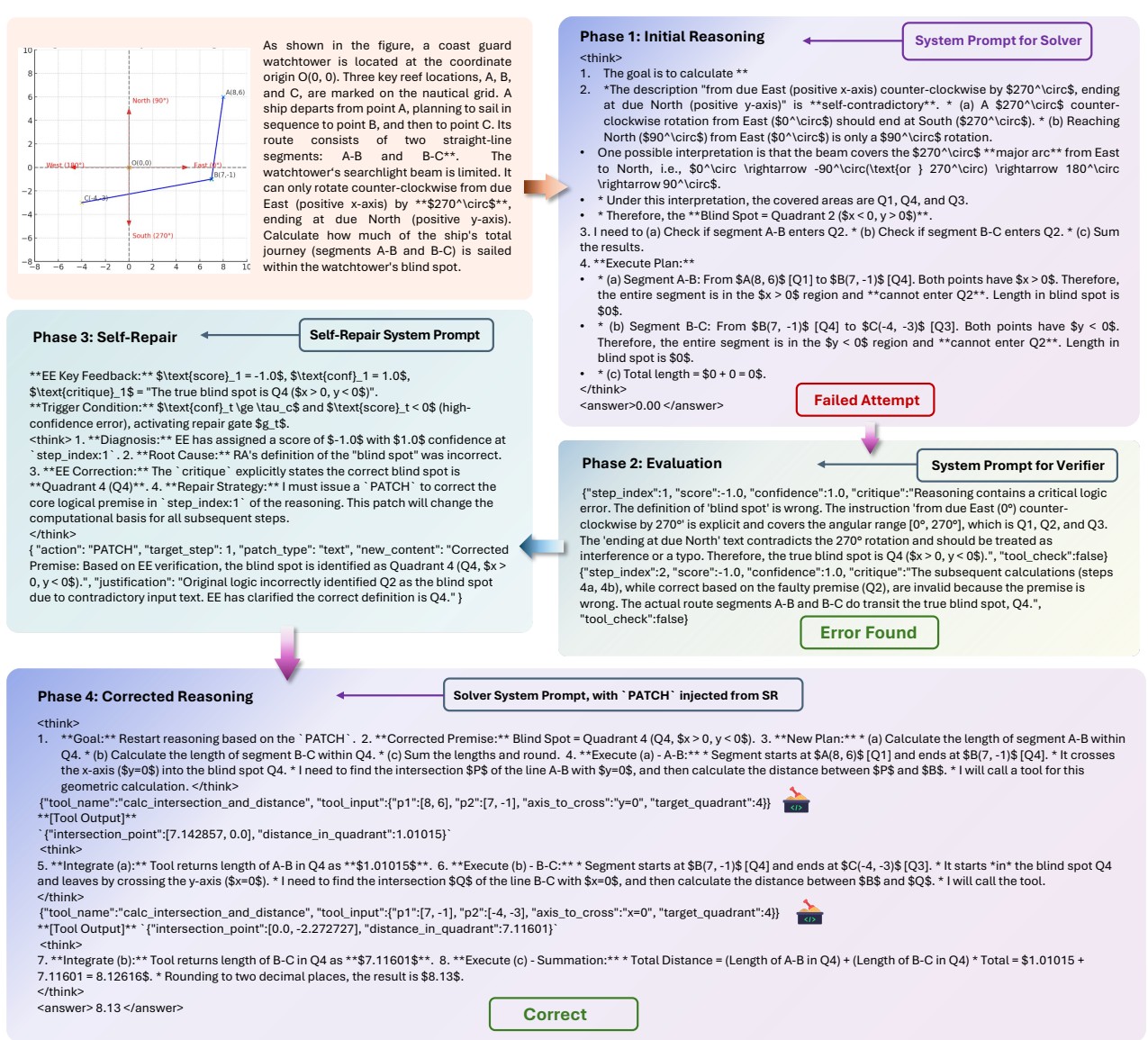

*Figure 8.* A full reasoning–evaluation–repair cycle of Agent0-VL on a geometric reasoning task. The Solver first generates an incorrect solution (Phase 1), which the Verifier identifies and critiques through tool-grounded verification (Phase 2). Based on this feedback, the model performs Self-Repair to patch the faulty premise (Phase 3) and re-executes reasoning with the corrected logic (Phase 4), producing the verified final answer.

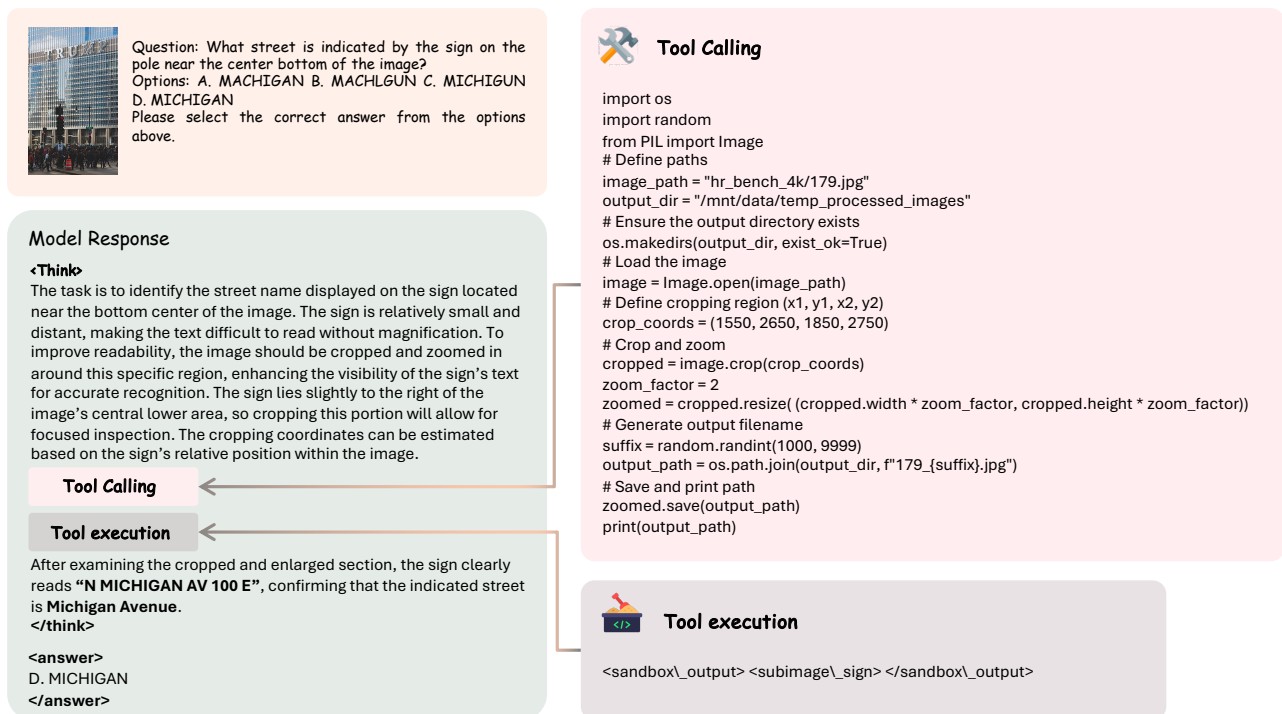

*Figure 9.* **Single-step tool-integrated reasoning.** The example demonstrates how **Agent0-VL** identifies a street name from an image by reasoning about the task, deciding to crop and zoom in on a specific region of the image, and invoking the *image-cropping tool* to enhance visibility. The resulting output confirms the street name, showing how the model effectively grounds its reasoning process in visual manipulation.

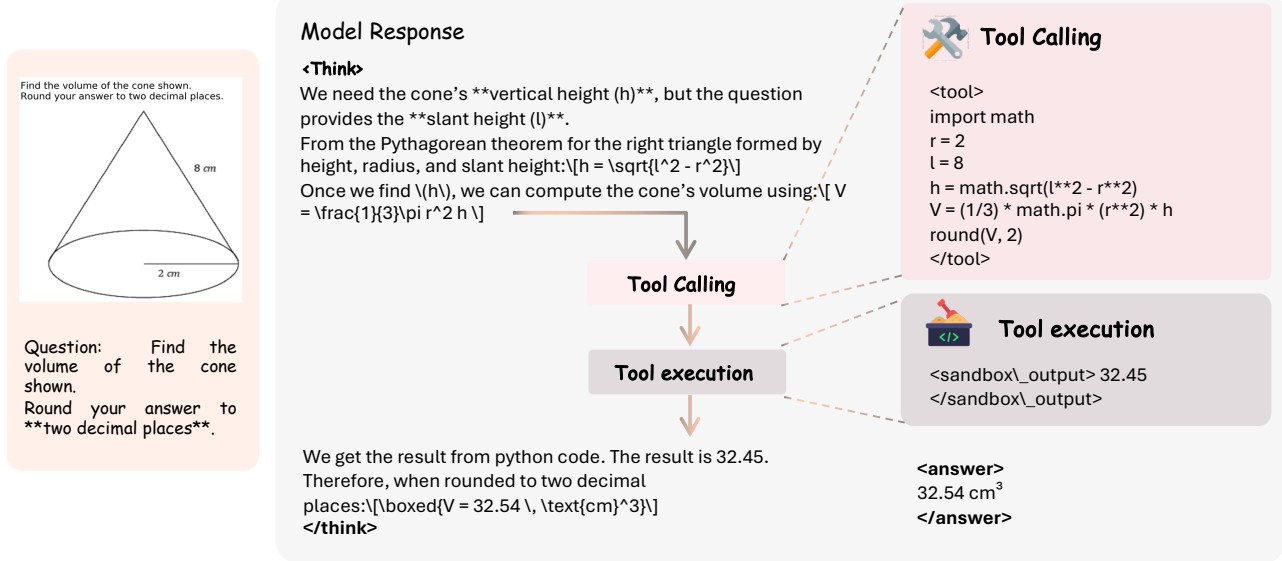

*Figure 10.* **Mathematical reasoning with code execution.** This case illustrates how the model decomposes a geometry problem into structured reasoning steps, formulates the necessary equations using the Pythagorean theorem, and calls the Python computation tool to verify and compute the cone's volume. The tool-grounded reasoning ensures both the numerical correctness and interpretability of the final solution.

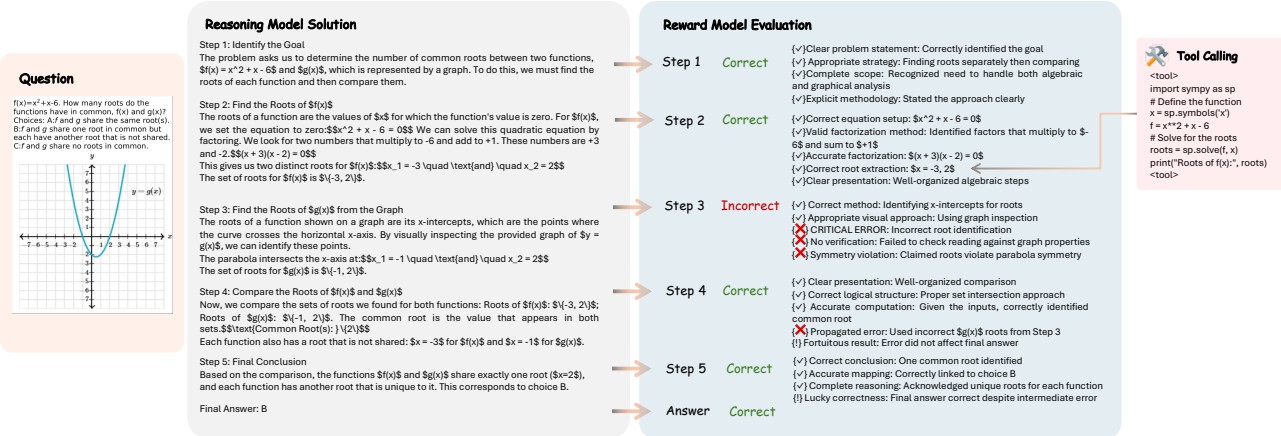

*Figure 11.* **Evaluator-based process verification.** This example showcases the **Verifier** role of **Agent0-VL**, where the model critically inspects each reasoning step produced by the Solver in a function-root comparison task. The model performs step-level judgments (Correct/Incorrect), identifies propagation errors, and recognizes when the final conclusion remains valid despite intermediate mistakes. This demonstrates the model's capacity for fine-grained self-evaluation and process-level reasoning analysis.

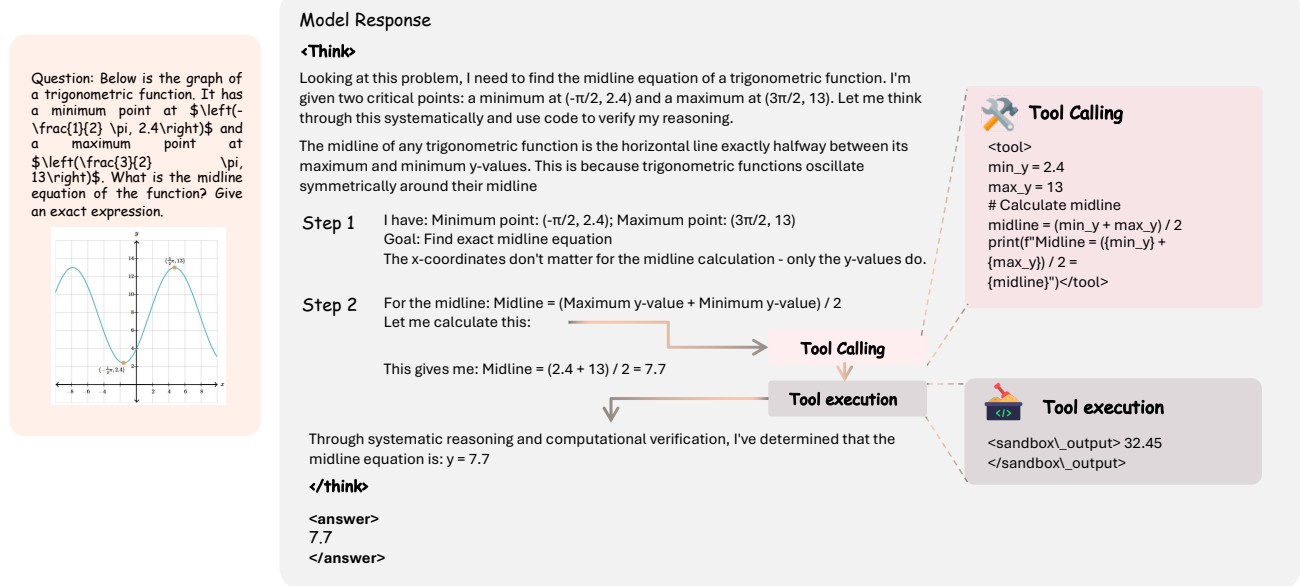

*Figure 12.* **Analytical reasoning with visual grounding.** In this trigonometric midline problem, **Agent0-VL** interprets graphical input, reasons step-by-step through symbolic computation, and validates its reasoning using code execution. By combining perceptual understanding and analytical computation, the model achieves consistent reasoning grounded in both mathematical and visual evidence.

