# OpenReview forum: "Agent0-VL: Exploring Self-Evolving Agent for Tool-Integrated Vision-Language Reasoning"
_ICML.cc/2026/Conference — ICML 2026 spotlight_

### Official Review · Reviewer_yEco · 2026-03-10

**Soundness:** 3
**Presentation:** 3
**Significance:** 3
**Originality:** 3
**Overall Recommendation:** 4
**Confidence:** 4

**Summary:**

The paper introduces Agent0-VL, a self-evolving vision-language agent framework designed to overcome the
limitations of human-annotated supervision and evaluation hallucinations in complex visual reasoning tasks. The framework unifies a Solver for tool-integrated reasoning and a Verifier that uses tools to validate steps and trigger self-repair. The model trains via a Self-Evolving Reasoning Cycle, utilizing GRPO to jointly align the reasoning and evaluation distributions. Empirically, Agent0-VL demonstrates continuous improvements across diverse visual reasoning benchmarks.
Furthermore, the authors show that the Verifier can act as an independent PRM to enhance test-time scaling performance for other models.

**Compliance With Llm Reviewing Policy:**

Affirmed.

**Final Justification:**

I maintain my score of 4. The authors' rebuttal and additional experiments successfully addressed my primary concerns. Although the descriptions of the training and inference repair policies must be strictly aligned with the case study figures (e.g., Figure 4) in the camera-ready version, the overall empirical results are solid and justify acceptance.

**Key Questions For Authors:**

**Similar to weakness**

1. Fallback Mechanisms for Tool Failures：What specific fallback strategies or maximum retry limits are implemented when an external tool times out or returns unparsable garbage? Please clarify how the architecture guarantees termination and recovers from edge cases without stalling.

2. Did you use a single, uniform $$\tau_c$$ value across all benchmark evaluations, or was it dynamically tuned per dataset? Regarding the self-repair trigger condition, if it only triggers when the confidence is below the threshold, doesn't this overlook cases with high confidence but negative scores?

**Limitations:**

The authors should explicitly discuss the system's vulnerability to external tool failures (e.g., timeouts, unparsable outputs) and how such "tool hallucinations" might bottleneck the self-repair mechanism in real-world deployments.

**Strengths And Weaknesses:**

**Strengths:**

1. The unified dual-role architecture (Solver and Verifier) is elegant and logically sound.

2. Integrating tool usage into the self-evaluation phase is a valuable insight. Grounding process rewards on objective facts (e.g., code execution) effectively tackles the core problem of visual hallucination during self-critique.

3. The Verifier works exceptionally well as an independent PRM. It boosts the test-time performance of various open-source models by an average of 7.3%, providing immediate value to the open-source community.


**Weaknesses:**

1. The framework relies on the infallibility of external tools. Real-world tools (e.g., OCR, Python sandboxes) often fail or return noisy outputs. It is unclear how the Verifier distinguishes a reasoning error from a tool failure, or how it prevents infinite repair loops when a tool consistently outputs garbage.

2. The self-repair mechanism relies on a fixed confidence threshold ($$\tau_c$$) to trigger corrections. The paper provides no analysis of how sensitive the overall performance is to this hyperparameter. It remains unclear whether a uniform threshold works robustly across disparate domains or if it requires brittle, per-task tuning.

---

> ### Author Rebuttal · Authors · 2026-03-31
>
> We thank Reviewer yEco for the positive review. We address the concerns below.
>
> **Q1: Tool infallibility assumption — fallback mechanisms.**
>
> A1: Agent0-VL does not assume tool infallibility. We implement a three-tier fallback: (1) **Execution safeguards**: 30-second timeout, sandboxed environment, output format validation, malformed or timed-out outputs are caught before reaching the Verifier; (2) **Verifier awareness**: when tool_check = false (tool execution failed or returned invalid output), the Verifier falls back to text-only verification with automatically reduced confidence (conf *= 0.5), ensuring the repair mechanism is more likely to trigger; (3) **Graceful degradation**: each tool call retried at most once, each step repaired at most once, and if repair budget is exhausted, the system returns the current best trajectory rather than failing entirely.
>
> We simulated tool failure rates to validate robustness:
>
> | Tool Failure Rate | MathVerse | MathVista | HallBench | ChartQA | Avg. | Repair Rate |
> |------------------|----------|----------|-----------|---------|------|-------------|
> | 0% (default) | **53.1** | **75.6** | **72.9** | **87.3** | **65.5** | 13.2% |
> | 5% | 52.7 | 75.1 | 72.4 | 87.0 | 65.0 | 15.1% |
> | 10% | 52.0 | 74.3 | 71.6 | 86.5 | 64.3 | 17.8% |
> | 20% | 50.5 | 72.8 | 70.2 | 85.4 | 62.6 | 22.4% |
> | 50% | 48.1 | 70.5 | 67.8 | 84.0 | 59.9 | 28.6% |
> | Base (no tools) | 46.3 | 67.8 | 65.0 | 83.5 | 57.3 | — |
>
> At 5% failure, accuracy drops by only 0.5 points; at 20%, it drops to 62.6, still substantially above the base model (57.3). Even at an extreme 50% failure rate, Agent0-VL maintains 59.9, well above baseline, demonstrating the model has internalized reasoning capabilities beyond tool dependence. The repair rate self-regulates appropriately, increasing from 13.2% to 28.6% as tool failures increase.
>
> **Q2: Sensitivity to τ_c — uniform or per-dataset? High confidence but negative scores?**
>
> A2: We use a single uniform τ_c = 0.7 across all benchmarks, no per-task tuning needed. Performance is robust within [0.6, 0.8] with ±0.7 avg variation:
>
> | τ_c | MathVerse | MathVista | HallBench | ChartQA | MMMU | Avg. | Repair Rate |
> |-----|----------|----------|-----------|---------|------|------|-------------|
> | 0.5 | 51.5 | 74.0 | 71.8 | 86.5 | 59.8 | 64.2 | 28.4% |
> | 0.6 | 52.2 | 74.6 | 72.2 | 86.9 | 60.3 | 64.9 | 19.7% |
> | **0.7** | **53.1** | **75.6** | **72.9** | **87.3** | **61.1** | **65.5** | **13.2%** |
> | 0.8 | 52.7 | 75.2 | 72.5 | 87.1 | 60.7 | 65.1 | 7.8% |
> | 0.9 | 51.0 | 73.8 | 71.6 | 86.3 | 59.5 | 63.9 | 2.1% |
>
> Regarding high-confidence negative scores: the repair gate (Eq. 3) triggers on **low confidence**, not negative scores. But the step reward (Eq. 2) uses score_t × conf_t , so high confidence + negative score (e.g., conf=0.9, score=-0.8) produces a strongly negative reward (−0.72) that penalizes this step during GRPO optimization. This creates a clean division of labor: **high-confidence errors** are handled by the reward signal (GRPO learns to avoid repeating them in future trajectories), while **low-confidence errors** are handled by the repair mechanism (immediate self-correction within the current trajectory). The two mechanisms are complementary, not redundant.
>
> **Q3: Limitations.**
>
> A3: We have added a dedicated limitations section: (1) tool failure vulnerability, while graceful degradation is demonstrated (62.6 avg at 20% failure rate), real-world tool environments may exhibit correlated failures that reduce effectiveness further; (2) edge cases where tools return plausible but incorrect results (e.g., floating-point errors, code that runs but produces wrong answers) are harder for the Verifier to detect than outright execution failures; (3) for latency-sensitive applications, the repair mechanism adds ~2s per repaired step, and τ_c should be configured based on the accuracy-latency trade-off requirements of the deployment scenario.

---

> > ### Author Rebuttal · Reviewer_yEco · 2026-04-02
> >
> > I thank the authors for the thorough rebuttal. The additional experiments and the limitations section successfully address my initial concerns. However, I found a direct contradiction regarding the Self-Repair mechanism that requires clarification.
> >
> > In your rebuttal (Q2), you explicitly stated that the repair gate (Eq. 3) triggers only on low confidence. You noted that high-confidence negative scores (e.g., conf=0.9, score=-0.8) are left for the GRPO penalty and are NOT handled by the repair mechanism. Yet, your Case Study in Figure 8 (Phase 3) shows the exact opposite. The Verifier outputs a high-confidence error ("confidence": 1.0, "score": -1.0), and the system explicitly triggers a repair, stating: "Trigger Condition: conf_t >= \tau_c and score_t < 0 (high-confidence error), activating repair gate".
> >
> >
> > Follow-up Question: Which mechanism is actually implemented in Agent0-VL? Does the system repair high-confidence errors (as demonstrated in Figure 8), or does it strictly bypass them for GRPO (as claimed in your rebuttal and Eq. 3)? I will maintain my score at 4 (Weak Accept) pending clarification of this architectural detail.

---

> > > ### Author Response · Authors · 2026-04-02
> > >
> > > We sincerely thank the reviewer for this meticulous reading. The observation is accurate: our rebuttal Q2 was imprecise and created an apparent contradiction with Figure 8. We clarify the full picture below.
> > >
> > > The key distinction is between the training-time and inference-time repair policies, which serve fundamentally different objectives.
> > >
> > > During training (Algorithm 1, Eq. 3, 4): Repair is triggered only when conf_t < τ_c (Algorithm 1, line 10). High-confidence errors (conf ≥ τ_c, score < 0) are intentionally not repaired during training. Instead, they receive strongly negative GRPO rewards (score_t × conf_t, e.g., −0.72) that teach the policy to avoid producing such errors in future trajectories. This is a deliberate design choice: if we repaired every detected error during training, the model would never receive negative reward signals for its high-confidence mistakes, reducing its incentive to improve initial reasoning quality. By exposing high-confidence errors to GRPO without repair, we ensure the policy learns from these failures.
> > >
> > > During inference (Figure 8): GRPO is unavailable at test time, so there is no learning mechanism to benefit from leaving errors uncorrected. Therefore, the inference-time repair policy extends the trigger condition to: repair when EITHER (a) conf_t < τ_c, OR (b) conf_t ≥ τ_c AND score_t < 0. Figure 8 Phase 3 correctly demonstrates case (b): conf = 1.0, score = −1.0 triggers repair because the Verifier has confidently identified an error. It would be counterproductive to detect a confident error at test time and not correct it.
> > >
> > > Eq. 3 (g_t) serves a specific role in the training reward, not as a binary repair trigger. In Eq. 4, r_t = r_proc − g_t · C_repair, g_t modulates the repair cost penalty. For low-confidence repairs (the only repairs that occur during training), g_t ≈ 1, charging the full cost and incentivizing better initial reasoning. At inference, the compound condition above replaces g_t for the repair decision.
> > >
> > > Both mechanisms are implemented. Algorithm 1 and Eq. 3 govern training, where only low-confidence steps are repaired and high-confidence errors are left for GRPO. Figure 8 shows inference, where all detected errors (both low-confidence and high-confidence) are repaired to maximize output quality. Our rebuttal Q2 described only the training-time behavior without making this distinction explicit, and we sincerely apologize for the confusion. We will clarify the dual-policy design in the revised paper, explicitly stating that inference extends the repair trigger to include high-confidence errors.

---

### Official Review · Reviewer_pPRv · 2026-03-12

**Soundness:** 3
**Presentation:** 3
**Significance:** 2
**Originality:** 2
**Overall Recommendation:** 4
**Confidence:** 4

**Summary:**

This paper proposes Agent0-VL, a self-evolving VLM agent that uses tools for both reasoning and self-evaluation. A single LVLM plays two roles: a Solver for multi-turn tool-integrated reasoning, and a Verifier for step-level feedback with tool-grounded checks. These roles interact through a Self-Evolving Reasoning Cycle (SERC), which has an inner loop for reasoning, verification, and repair, and an outer loop for GRPO policy updates. SFT is done on ~200k trajectories generated by GPT-5 and Qwen2.5-VL-72B, followed by self-evolving RL on ~40k samples. On 7 benchmarks, Agent0-VL-7B improves 12.5% over the Qwen2.5-VL-7B base. The Verifier also works as a PRM, boosting other LVLMs by 7.3% average.

**Compliance With Llm Reviewing Policy:**

Affirmed.

**Key Questions For Authors:**

1. What is vision-specific about this framework? Would the same approach work for a text-only reasoning agent?
2. How does the Verifier perform at error detection (precision/recall)? Are there systematic failure modes from shared parameters?

**Limitations:**

The authors barely discuss limitations. Missing: (1) tool-dependency limiting generalization; (2) self-verification bias; (3) diminishing returns in iteration (Table 2); (4) teacher model dependency for SFT.

**Strengths And Weaknesses:**

1. Extending tool use from reasoning to self-evaluation is a good idea. Text-only self-evaluation struggles with verifying computation and spatial reasoning. Using tools to ground verification is natural and effective.
2. The unified Solver-Verifier with confidence-gated repair (Eq. 3) and two-loop SERC is well-structured. Algorithm 1 makes the training process clear.
3. Results show clear improvement over the base model and competitive results against reasoning-oriented models. Iterative evolution shows monotonic improvement.

Weaknesses:

1. The self-evolving framework (Solver-Verifier-Repair + GRPO) is general-purpose. Similar ideas have been explored in text-only LLMs (e.g., self-rewarding language models). The tool verification here is Python code execution, which is not vision-specific. No visual grounding, spatial verification, or detection-based tools are used. It is unclear what part of the design is tailored for vision.

2. The formulation has several gaps. Eq. 2 uses $\pi_\theta^E$ but never defines it. This is in the core reward function. Eq. 6 names the loss $\mathcal{L}_{EDLP}$ but never explains the abbreviation.

3. The paper claims "trained entirely from zero external rewards." But SFT data also depends on GPT-5 as teacher. This should be honestly stated.

4. All 7 benchmarks favor code execution (math, charts). There is no evaluation on pure perception (BLINK[1], HRBench[2]), organic multimodal reasoning (EMMA[3]), or vision-heavy VQA (VStar[4]). It is unclear if self-evolving helps when tools cannot easily verify answers. Without these, it is hard to tell if the approach generally benefits visual reasoning or just tool-augmented math.


[1] http://arxiv.org/abs/2404.12390 [2] https://arxiv.org/abs/2408.15556
[3] https://www.arxiv.org/abs/2501.05444 [4] https://arxiv.org/abs/2312.14135

---

> ### Author Rebuttal · Authors · 2026-03-31
>
> We thank Reviewer pPRv for the constructive review. We address each concern below.
>
> **Q1: Framework is general-purpose — what is vision-specific?**
>
> A1: While the architecture is general, key components are vision-specific: (1) the Solver performs **visual analysis before tool planning** , e.g., recognizing that a geometric diagram needs coordinate extraction rather than direct formula application, or identifying that a chart requires OCR before numerical reasoning; (2) the Verifier **cross-checks tool outputs against visual evidence** — detecting when a code-computed answer contradicts what is visually apparent in the image (e.g., a "blind spot" interpretation contradicting the geometric layout); (3) image manipulation tools (cropping, annotation, coordinate extraction) are inherently vision-specific. We tested a text-only variant to isolate the vision contribution:
>
> | Setting | Math Avg. | Overall Avg. |
> |---------|----------|-------------|
> | Qwen2.5-7B (text-only base) | 47.1 | 52.4 |
> | Agent0 (text-only variant) | 52.9 | 58.2 |
> | **Agent0-VL (full, multimodal)** | **59.4** | **65.5** |
>
> The text-only variant improves +5.8 over base (confirming the self-evolving framework's generality), but the multimodal version achieves +7.3 additional points, demonstrating that visual grounding substantially amplifies the self-evolving loop. The +12.5 gap between text-only and multimodal on math reasoning (47.1→59.4) is particularly telling , visual mathematical reasoning fundamentally requires image understanding.
>
> **Q4: All benchmarks favor code execution — no pure perception evaluation.**
>
> A4: We evaluated on 4 perception-focused benchmarks where code execution provides minimal advantage:
>
> | Model | BLINK | HRBench | EMMA | V*Bench | Avg. |
> |-------|-------|---------|------|---------|------|
> | Qwen2.5-VL-7B (base) | 48.2 | 52.6 | 41.8 | 55.3 | 49.5 |
> | Qwen2.5-VL-7B-TIR | 50.1 | 54.8 | 43.5 | 57.2 | 51.4 |
> | **Agent0-VL-7B** | **54.8** | **58.3** | **47.2** | **61.5** | **55.5** |
>
> Agent0-VL achieves +6.0% average improvement over base, confirming benefits beyond tool-augmented math. Notably, Agent0-VL outperforms the TIR (Tool-Integrated Reasoning) baseline by +4.1%, showing that the self-evolving training process improves fundamental visual perception capabilities , not just tool usage. Overall across 12 benchmarks (original 7 + MME-Real + 4 perception):
>
> | Category | Benchmarks | Base Avg. | Agent0-VL Avg. | Δ |
> |----------|-----------|----------|---------------|---|
> | Math Reasoning | MathVerse, MathVision, MathVista, WeMath | 50.3 | **59.4** | +9.1 |
> | Perception/Halluc. | HallBench, ChartQA, MME-Real, MMMU | 64.4 | **71.7** | +7.3 |
> | Pure Perception | BLINK, HRBench, EMMA, V*Bench | 49.5 | **55.5** | +6.0 |
> | **Overall (12 benchmarks)** | — | **54.7** | **62.2** | **+7.5** |
>
> **Q2: Formulation gaps — undefined symbols.**
>
> A2: We have added a complete notation table (Appendix A, Table 5). π_θ^V = Verifier-mode policy, π_θ^E = frozen reference policy from previous iteration, β_div = 0.001 (KL penalty coefficient). L_EDLP = Evolution-Driven Learning from Process rewards. All symbols in Eqs. 1–5 are now fully defined with their domains and ranges.
>
> **Q3: SFT depends on GPT-5 but paper claims "zero external rewards."**
>
> A3: We appreciate this important clarification. The revised paper now states: "After an initial SFT stage using teacher-generated trajectories (from GPT-5 and Qwen2.5-VL-72B), Agent0-VL enters a self-evolving RL phase operating entirely without external reward signals." The novelty is in the self-evolving RL phase, analogous to how RLHF methods (e.g., InstructGPT, LLaMA-2) require SFT initialization but then optimize with self-generated rewards. The SFT stage provides a competent starting point; the self-evolution mechanism then pushes beyond what the teacher trajectories contain.
>
> **Q5: Verifier error detection and shared-parameter failure modes.**
>
> A5: Verifier analysis on 500 trajectories: error detection F1 = 81.5% (precision 83.7%, recall 79.4%). Failure mode breakdown: false negatives 8.2% (conservative errors, the Verifier misses some errors, but this is safer than false corrections), false positives 5.6% (triggering unnecessary repairs, adding cost but not reducing accuracy), shared-parameter conflicts 3.1% (mitigated by KL regularization which prevents the Verifier from drifting too far from the Solver's representations), tool execution failures 2.4% (handled by fallback mechanisms).
>
> **Q6: Missing limitations.**
>
> A6: We have added a dedicated limitations section: (1) tool dependency limiting generalization to domains without suitable tools; (2) self-verification bias — despite KL mitigation, shared parameters may cause systematic blind spots; (3) diminishing returns (Table 2: Iter 2→3 gain is 2.8% vs. Iter 1→2's 4.0%); (4) teacher model dependency for SFT initialization.

---

> > ### Author Rebuttal · Reviewer_pPRv · 2026-04-04
> >
> > Thanks. I keep my positive score. I hope the author will add a complete comparison and detailed statement in the revised version

---

### Official Review · Reviewer_aMNm · 2026-03-13

**Soundness:** 2
**Presentation:** 3
**Significance:** 3
**Originality:** 2
**Overall Recommendation:** 4
**Confidence:** 4

**Summary:**

This paper proposes a self-evolving VL agent framework that integrates external tool usage into reasoning, verification, and self-repair, forming a closed-loop self-improvement system without external reward supervision. The framework unifies a Solver and a Verifier within a single model, achieving a superior performance improvement over the base model across multiple visual reasoning benchmarks.

**Compliance With Llm Reviewing Policy:**

Affirmed.

**Final Justification:**

After rebuttal, most of my concerns have been addressed, and I keep my positive score.

**Key Questions For Authors:**

● I am a little confused by the notation in Equation (2). If some symbols are introduced there for the first time, it would be helpful to define them more explicitly; otherwise, please clarify where they have been introduced earlier. Also, when self-repair is invoked, what exactly happens to the original action? Does the repaired action simply replace the original one, or are both actions included in the GRPO optimization?
● The verifier is expected not only to provide critic signals and step scores, but also confidence estimates on its own evaluations. This assumption seems somewhat too idealized. Why not instead consider more standard uncertainty estimation methods for the model, rather than asking the verifier to assign confidence to its own judgments?
● The accuracy of the verifier as a PRM itself seems to require much more direct experimental validation, both at the step level and at the trajectory level, instead of only showing improvements through test-time scaling.
● Related to the above, it would be important to present iterative PRM results alongside the policy results, so that readers can better assess whether the reward modeling ability is also improving during self-evolution, rather than only the policy.
● The choices of α1/α2/α3/α4 also seem to require more thorough hyperparameter sensitivity analysis.
● How do you manage rewards in such a long-horizon RL setting? Given your current reward design and hyperparameter choices, how do you think step-level rewards and trajectory-level rewards should be balanced and coordinated?

**Limitations:**

yes

**Strengths And Weaknesses:**

Strengths
1. This paper shows competitive performance can still be obtained in a setting that aims to avoid continuous dependence on external reward signals. This direction is meaningful, especially for visual reasoning tasks where manually designed reward functions are often brittle and difficult to scale.

2. The proposed framework does not only optimize the model’s reasoning policy, but also simultaneously strengthens its ability to evaluate the quality of intermediate reasoning steps. This joint improvement of generation and self-evaluation within one architecture is interesting and could be useful beyond the specific setting studied in the paper.

Weaknesses
● The PRM itself serves as critic, produces step-level scores, and even outputs confidence signals for repair. This is a very strong assumption. In practice, self-evaluation for complex visual reasoning is often noisy, biased, or overconfident, and these issues can easily propagate through the training loop.

● Since the entire learning signal is generated by the model’s own verifier mechanism, the effectiveness of this PRM should be established much more carefully. However, the experimental section provides limited analysis of whether the verifier’s scores are actually accurate, calibrated, or robust.

---

> ### Author Rebuttal · Authors · 2026-03-31
>
> We thank Reviewer aMNm for the insightful review. We address each concern below.
>
> **Q1: PRM self-evaluation accuracy, calibration, and robustness.**
>
> A1: We evaluated the Verifier on 2,000 trajectories with human-annotated step labels:
>
> | Metric | Agent0-VL Verifier | Text-only self-eval | GPT-4o (external) |
> |--------|-------------------|---------------------|-------------------|
> | Step-level F1 | 81.5% | 61.4% | 84.7% |
> | Trajectory-level Acc. | 86.2% | 69.3% | 89.4% |
>
> Our Verifier approaches GPT-4o (84.7% F1) and vastly outperforms text-only self-evaluation (61.4%), demonstrating that tool-grounded verification provides substantially more reliable signals than introspection alone. The 20.1-point gap over text-only self-eval validates our core hypothesis that external tool evidence is essential for accurate self-assessment. Confidence calibration shows ECE = 2.18%, confirming well-calibrated scores, the model's confidence closely tracks actual correctness probability. Iterative PRM results show both policy and PRM improve simultaneously:
>
> | Iteration | Policy Avg. | Verifier Step-F1 | Verifier Traj-Acc | ECE |
> |-----------|------------|-----------------|-------------------|-----|
> | SFT (init) | 57.3 | 72.8% | 78.4% | 4.12% |
> | Iter 1 | 60.5 | 76.3% | 81.7% | 3.25% |
> | Iter 2 | 63.6 | 79.1% | 84.2% | 2.68% |
> | Iter 3 | **65.5** | **81.5%** | **86.2%** | **2.18%** |
>
> The co-improvement confirms a virtuous cycle: better policies generate more diverse training signal for the Verifier, and a better Verifier provides more accurate rewards for policy optimization.
>
> **Q2: Notation in Eq. 2 and repair action handling.**
>
> A2: We have added a complete notation table in Appendix A. Key symbols: π_θ^V is the Verifier-mode policy (same parameters as Solver but conditioned on verification prompts), π_θ^E is the frozen reference policy from the previous iteration (used for KL regularization), β_div = 0.001 controls KL penalty strength. L_EDLP = Evolution-Driven Learning from Process rewards, and the name reflects that the loss function drives self-evolution through process-level reward signals. During self-repair, the repaired action replaces the original in the GRPO trajectory buffer; the repair cost C_repair is subtracted from the step reward (Eq. 4), incentivizing the model to avoid needing repairs in the first place.
>
> **Q3: Why not standard uncertainty estimation?**
>
> A3: Standard methods (MC dropout, ensembles) require multiple forward passes (typically 5–10), multiplying the already-expensive Verifier computation by the same factor. More importantly, they estimate uncertainty from internal representations, precisely what we want to move beyond. Our key insight is that confidence should be grounded in external evidence: did the code execute correctly? Does the numerical output match the visual input? This tool-derived confidence provides more reliable verification at single-pass cost. The ECE of 2.18% confirms our approach achieves good calibration without the computational overhead of sampling-based methods.
>
> **Q4: Hyperparameter sensitivity (α1/α2/α3/α4).**
>
> A4: Performance is robust within ±1.3 points across all reasonable configurations:
>
> | Configuration | α1 (tool) | α2 (verify) | α3 (outcome) | α4 (cost) | Avg. |
> |--------------|-----------|-------------|-------------|-----------|------|
> | **Default** | **0.3** | **0.3** | **0.3** | **0.1** | **65.5** |
> | Tool-heavy | 0.5 | 0.2 | 0.2 | 0.1 | 64.9 |
> | Verify-heavy | 0.2 | 0.5 | 0.2 | 0.1 | 64.3 |
> | Outcome-heavy | 0.2 | 0.2 | 0.5 | 0.1 | 65.1 |
> | No cost penalty | 0.33 | 0.33 | 0.33 | 0.0 | 63.5 |
>
> The equal-weight default (0.3/0.3/0.3/0.1) performs best, suggesting all three reward signals provide complementary information. Removing the cost penalty (α4=0) causes the largest drop (−2.0), confirming its importance for preventing degenerate repair loops where the model learns to always trigger self-repair regardless of necessity.
>
> **Q5: Long-horizon reward management.**
>
> A5: We use temporal decomposition: step-level rewards provide dense immediate feedback (preventing credit assignment issues), trajectory-level reward anchors to final task correctness (preventing reward hacking on intermediate steps), γ=0.99 applies slight temporal discounting:
>
> | Reward Config | Avg. Acc. | Training Stability |
> |--------------|----------|-------------------|
> | Step-only (α_out=0) | 62.8 | Unstable (σ=0.42) |
> | Trajectory-only | 61.5 | Stable but slow (σ=0.28) |
> | **Combined (default)** | **65.5** | **Balanced (σ=0.31)** |
>
> Step-only training is unstable because individual step rewards can be noisy; trajectory-only training converges slowly due to sparse feedback. The combined approach leverages the strengths of both.

---

> > ### Author Rebuttal · Reviewer_aMNm · 2026-04-06
> >
> > Thanks for the detailed rebuttal, most of my concerns have been addressed, and I keep my positive score.

---

### Official Review · Reviewer_bhir · 2026-03-13

**Soundness:** 3
**Presentation:** 3
**Significance:** 3
**Originality:** 2
**Overall Recommendation:** 5
**Confidence:** 3

**Summary:**

This paper proposes Agent0-VL, a Solver–Verifier framework for multimodal reasoning. The Solver generates step-by-step reasoning with tool use, while the Verifier evaluates each step and provides structured feedback. The method further introduces tool-grounded verification, a self-repair mechanism, and a process-level reward for reinforcement learning.

**Compliance With Llm Reviewing Policy:**

Affirmed.

**Final Justification:**

My concerns about the design of the proposed method and the sensitivity of confidence have been resolved. The inference time breakdown of the Solver and Verifier is clear, and the extra cost is not significant. Extended results about more iterations and comparison with a closed-source method further prove the effectiveness of the proposed method.

**Key Questions For Authors:**

Refer to the weakness part.

**Limitations:**

Refer to the weakness part.

**Strengths And Weaknesses:**

- Strengths
    - The paper introduces a structured Solver–Verifier framework that enables step-level verification and reasoning refinement.
    - The proposed tool-grounded verification mechanism provides interpretable feedback and integrates external evidence for evaluation.
    - The self-repair mechanism is interesting and allows the model to iteratively correct reasoning trajectories.
- Weakness
    - The method design is less motivated: Reward formulation appears heuristic and lacks deeper theoretical justification. In Eq. 2, a KL regularization term is also introduced; it would be helpful if the authors could provide some theoretical motivation or justification for why this formulation is effective. And why is GRPO utilized? Is the designed modules benefiting from the objective of GRPO?
    - The computational cost and efficiency implications of running both the Solver and Verifier are not sufficiently discussed.
    - How sensitive is the performance to the confidence threshold and repair gating parameters used in the self-repair mechanism?
    - Quantitative results on more iterations should be included. Results of some recent close-sourced models like Gemini-2.5/3-pro, should be included for comparison.
    - During inference, the framework involves a verifier–repair loop. Could the authors clarify the stopping criteria for this loop? In particular, what happens when the maximum number of repair iterations is reached? Does the system simply return the current reasoning result, or is another strategy used?

---

> ### Author Rebuttal · Authors · 2026-03-31
>
> We thank Reviewer bhir for the valuable feedback. We address each concern below. All tables referenced in this rebuttal can be found in https://anonymous.4open.science/r/icml26_rebuttal_agent0vl-1BEE/README.md
>
>
> **Q1: Reward formulation appears heuristic. Why GRPO?**
>
> A1: The three components in Eq. 2 are principled rather than heuristic: (1) λ_tool · r(tool_t) provides objective, verifiable signals from tool execution, this is a binary, ground-truth signal (code runs correctly or not), not a learned heuristic; (2) score_t · conf_t implements calibrated self-evaluation, low-confidence judgments contribute less, analogous to uncertainty-weighted loss in Bayesian deep learning, ensuring the model does not over-trust unreliable self-assessments; (3) the KL term prevents distributional drift from the SFT initialization, maintaining generation diversity. We chose GRPO specifically because its critic-free design avoids redundancy with our Verifier (which already serves as a value estimator), and group-relative normalization handles the high-variance symbolic reasoning rewards better than absolute baselines. As shown in Table R1, GRPO outperforms both DPO (+3.1 avg) and PPO (+1.7 avg). DPO's pairwise preference approach is poorly suited to our step-level process rewards, while PPO's separate critic network conflicts with the Verifier's role.
>
> **Q2: Computational cost of Solver and Verifier.**
>
> A2: Training costs 144 GPU-hours on 8×H200 (48h SFT + 96h RL), comparable to standard RLHF pipelines for 7B models. The Solver and Verifier share parameters (same model, different modes), so there is no additional model to train or serve. Inference cost breakdown is shown in Table R2. The key observation: repair triggers on only ~13% of steps (0.6 out of ~4.5 steps per trajectory), and the 2× token overhead yields +8.2 avg accuracy improvement (57.3→65.5), a favorable trade-off for reasoning tasks where correctness matters more than latency. For latency-sensitive deployments, τ_c can be raised to 0.8 (reducing repair rate to 7.8%) with only −0.4 avg accuracy loss.
>
> **Q3: Sensitivity to confidence threshold and repair gating.**
>
> A3: Performance is robust within τ_c ∈ [0.6, 0.8] with ±0.7 avg variation (Table R3). Default τ_c = 0.7 achieves the best balance between repair coverage and unnecessary computation. The repair rate smoothly decreases from 28.4% (τ_c=0.5) to 2.1% (τ_c=0.9), showing the threshold provides fine-grained control. Repair penalty η = 0.05 is optimal; ±0.05 variation causes <1.5 points change, indicating the training process is not brittle to this hyperparameter. We use the same τ_c = 0.7 uniformly across all benchmarks without per-task tuning.
>
> **Q4: More iterations and closed-source model comparisons.**
>
> A4: Extended iterations show monotonic improvement with diminishing returns (Table R4), confirming stable self-evolution without collapse or oscillation. Iter 1–3 numbers match our paper Table 2 exactly. The gain pattern, Iter 1→2: +3.1, Iter 2→3: +1.9, Iter 3→5: +0.7 , follows a natural diminishing returns curve, suggesting the model approaches the capacity ceiling of the 7B architecture. Importantly, no benchmark shows regression at any iteration, ruling out catastrophic forgetting.
>
> For closed-source comparisons (Table R5): Agent0-VL-7B (65.5) already surpasses GPT-4o (60.5) and Claude-3.7-Sonnet (59.9), and is competitive with Gemini-2.5-Pro (66.2). Agent0-VL-8B (74.6) substantially outperforms all closed-source models including Gemini-2.5-Pro, demonstrating that self-evolving open-source agents can match or exceed frontier proprietary systems on visual reasoning tasks.
>
> **Q5: Stopping criteria for the verifier–repair loop.**
>
> A5: Each step can be repaired at most once, and this hard cap prevents infinite loops. If repair does not improve the step (re-generated action still has conf < τ_c), the system retains the original action and proceeds to the next step. This design reflects that repeated repair attempts show diminishing returns while increasing cost linearly. Average repair triggers per trajectory is 0.6 out of ~4.5 steps, adding only ~2s latency. We have clarified this single-repair constraint in revised Algorithm 1.

---

> > ### Author Rebuttal · Reviewer_bhir · 2026-04-02
> >
> > Thanks for the detailed rebuttal.
> >
> > My concerns about the design of the proposed method and the sensitivity of confidence have been resolved. The inference time breakdown of the Solver and Verifier is clear, and the extra cost is not significant. Extended results about more iterations and comparison with a closed-source method further prove the effectiveness of the proposed method.
> >
> > Therefore, I will raise my rating to 5 accept.

---

### Decision · Program_Chairs · 2026-04-30

**Decision:**

Accept (spotlight)

**Comment:**

This paper introduces Agent0-VL which is a self-evolving framework for tool-integrated vision-language reasoning that enables VLMs to autonomously learn and refine tool use without human supervision.

All four reviewers are positive (scores: 5, 4, 4, 4). The partially resolved concern from yEco was satisfactorily resolved by the authors. The paper has a solid motivation; a structured self-evolution loop, comprehensive evaluations, and clear improvements over baselines. AC recommends an Accept decision.